# Dysregulation of Mitochondrial Function in Cancer Cells

**DOI:** 10.3390/ijms26146750

**Published:** 2025-07-14

**Authors:** Ahmed Mahmoud Ahmed Mahmoud Awad, Norwahidah Abdul Karim

**Affiliations:** Department of Biochemistry, Faculty of Medicine, Universiti Kebangsaan Malaysia, Kuala Lumpur 56000, Malaysia; p123165@siswa.ukm.edu.my

**Keywords:** mitochondria, cancer metabolism, bioenergetics, mitochondrial dynamics, OXPHOS

## Abstract

In addition to their well-known role in ATP production, mitochondria are vital to cancer cell metabolism due to their involvement in redox regulation, apoptosis, calcium signaling, and biosynthesis. This review explores how cancer cells drive the extensive reprogramming of mitochondrial structure and function, enabling malignant cells to survive hostile microenvironments, evade therapy, and proliferate rapidly. While glycolysis (the Warburg effect) was once thought to be the dominant force behind cancer metabolism, recent updates underscore the pivotal contribution of mitochondrial oxidative phosphorylation (OXPHOS) to tumor development. Cancer cells often exhibit enhanced mitochondrial ATP production, metabolic flexibility, and the ability to switch between energy sources such as glucose, glutamine, and pyruvate. Equally important are changes in mitochondrial morphology and dynamics. Due to disruptions in fusion and fission processes, regulated by proteins like Drp1 and MFN1/2, cancer cells often display fragmented mitochondria, which are linked to increased motility, metastasis, and tumor progression. Moreover, structural mitochondrial alterations not only contribute to drug resistance but may also serve as biomarkers for therapeutic response. Emerging evidence also points to the influence of oncometabolites and retrograde signaling in reshaping mitochondrial behavior under oncogenic stress. Collectively, these insights position mitochondria as central regulators of cancer biology and attractive targets for therapy. By unraveling the molecular mechanisms underlying mitochondrial reprogramming—from energy production to structural remodeling—researchers can identify new approaches to disrupt cancer metabolism and enhance treatment efficacy.

## 1. Introduction

Mitochondria are central to cellular energy metabolism, generating most of cellular ATP through oxidative phosphorylation (OXPHOS). In addition to energy production, these organelles play pivotal roles in regulating reactive oxygen species (ROS), apoptosis, calcium signaling, and various biosynthetic pathways, underscoring their essential role in maintaining cellular homeostasis. In cancer, mitochondrial function is often profoundly altered, reflecting the metabolic reprogramming that supports rapid cell proliferation, survival under stress, and resistance to therapy [1].

Traditionally, cancer metabolism was characterized by the Warburg effect, which suggested that cancer cells favor glycolysis for ATP production even in the presence of oxygen [2]. However, recent findings reveal that mitochondrial metabolism remains active and crucial in tumor growth and progression. Mitochondria in cancer cells exhibit remarkable metabolic plasticity and adaptability, enabling them to switch between various fuel sources, such as glucose, fatty acids, and glutamine, to meet energy and biosynthetic demands. This metabolic flexibility allows tumor cells to thrive under metabolic stress and fluctuating nutrient availability [2].

Proteomic and functional analyses across various cancer cell lines have uncovered distinct mitochondrial alterations, including enhanced spare respiratory capacity, increased basal respiration, and elevated ATP-linked respiration. These adaptations reflect the heightened bioenergetic and biosynthetic demands of cancer cells [3]. Moreover, disruptions in mitochondrial dynamics, such as imbalances in fusion and fission, along with mutations in mitochondrial DNA (mtDNA), have been associated with oncogenesis and cancer progression. Structural and functional abnormalities in mitochondria are also implicated in altered redox signaling, fostering a pro-survival environment that supports malignant cell growth [4].

Notably, reports from metabolic pathway profiling and real-time ATP production assays have shown that several types of cancers are highly dependent on mitochondrial ATP production. This dependence is particularly pronounced in cells that rely on oxidative metabolism for their growth and survival. For example, breast cancer cells with impaired mitochondrial function exhibit reduced tumorigenicity potential, suggesting the therapeutic promise of targeting mitochondrial pathways [5]. Additionally, tumor cells can manipulate the surrounding stromal cells to supply mitochondrial fuels like lactate and ketone bodies, further emphasizing the complexity of the tumor metabolic network [6].

As the understanding of cancer metabolism evolves, mitochondria emerge not only as key players in tumor biology but also as promising therapeutic targets. Investigating alterations in mitochondrial bioenergetics, fuel flexibility, and proteomic profiles in cancer cells can uncover the vulnerabilities that may be leveraged as targets for anti-cancer therapies. This review explores the multifaceted roles of mitochondria in cancer cells, highlighting their functional adaptations, metabolic reprogramming, and potential as targets for therapeutic interventions.

## 2. Alteration of Mitochondrial Function in Cancer Cells

In healthy cells, mitochondria not only provide the majority of cellular energy but also regulate calcium homeostasis, redox signaling, and programmed cell death. In cancer cells, however, mitochondrial function is often reprogrammed to support the increased bioenergetic and biosynthetic demands of uncontrolled proliferation. Emerging evidence suggests that mitochondrial dysfunction, coupled with changes in mitochondrial morphology and dynamics, plays a critical role in tumor development and progression [7]. The central role of mitochondria in energy metabolism is facilitated by the electron transport chain (ETC), which couples the transfer of electrons from NADH and FADH_2_ to oxygen with the generation of a proton gradient across the inner mitochondrial membrane. This electrochemical gradient drives ATP synthesis via ATP synthase (complex V). Despite the classic view stemming from the Warburg hypothesis that cancer cells primarily rely on aerobic glycolysis, recent findings have demonstrated that mitochondria remain functional in many cancers and contribute significantly to cellular ATP pools. In some tumors, mitochondrial ATP production even surpasses that of glycolysis, underscoring the persistent importance of oxidative phosphorylation in cancer metabolism [8]. A study investigating the effects of K-Ras oncogene activation in mouse fibroblasts demonstrated that transformed cells exhibited significant alterations in mitochondrial gene expression and morphology compared to their non-transformed counterparts. Transformed fibroblasts showed decreased ATP levels, reduced capacity to adapt mitochondrial morphology in response to glucose depletion, and partial mitochondrial uncoupling. These mitochondrial defects were directly linked to oncogenic Ras activation and were reversible upon the expression of a dominant-negative GEF, highlighting a causal relationship between oncogene signaling and mitochondrial dysfunction. This finding suggests that in addition to glycolytic upregulation, mitochondrial impairment may be a direct consequence of oncogenic pathways, particularly those involving Ras family proteins [9]. In another human model of mitochondrial dysfunction, scholars investigated fibroblasts derived from Parkinson’s disease patients carrying mutations in the Parkin gene [6]. Although not derived from cancer cells, this model provides key insights into how mitochondrial abnormalities manifest at the functional and morphological levels in disease states. The mutant cells demonstrated significantly lower ATP production, increased oxidative stress, and altered mitochondrial membrane potential under stress conditions. Morphological analysis revealed reduced mitochondrial branching following oxidative stress, indicating compromised mitochondrial dynamics. These findings parallel alterations often observed in cancer cells, such as diminished OXPHOS efficiency and fragmented mitochondrial networks, which may be adaptive responses to stress or reflect impaired mitophagy and quality control mechanisms [6].

A critical aspect of mitochondrial dysfunction in cancer is the disruption of mitochondrial dynamics: the processes of mitochondrial fusion and fission that regulate mitochondrial shape, distribution, and function. A comprehensive study examined the impact of mitochondrial DNA (mtDNA) depletion and oxidative stress on mitochondrial dynamics and retrograde signaling [10]. The results showed that cells with depleted mtDNA had disorganized cristae, reduced mitochondrial membrane potential, and an increased reliance on fission machinery such as DRP1. Simultaneously, mitochondrial fusion proteins such as OPA1 were downregulated. These structural changes correlated with a migratory phenotype and increased expression of stress response gene features commonly associated with tumor progression [10]. Importantly, the study linked mitochondrial dysfunction to the activation of retrograde signaling pathways, including calcium–calcineurin signaling and oxidative stress-mediated transcriptional programs. This signaling crosstalk between mitochondria and the nucleus enables a cell to adapt its gene expression in response to mitochondrial damage but may also facilitate tumor survival and metastasis. For instance, increased mitochondrial ROS production can activate AMPK or stabilize HIF-1α, promoting angiogenesis, glycolysis, and resistance to apoptosis, traits that enhance tumor cell fitness in hypoxic environments [10]. Moreover, the specific genetic alterations in mitochondrial enzymes have been implicated in hereditary and sporadic cancers. Mutations in components of complex II (succinate dehydrogenase) and fumarate hydratase impair the tricarboxylic acid (TCA) cycle and contribute to the accumulation of oncometabolites such as succinate and fumarate. These metabolites inhibit prolyl hydroxylases, stabilizing HIF-1α and promoting tumor angiogenesis and growth. Additional studies have identified mutations in complex I subunits that enhance ROS production and activate pro-survival signaling cascades [8,9,10]. Collectively, these findings challenge the outdated notion that mitochondria are largely irrelevant in cancer cell metabolism. Instead, they reveal that mitochondrial dysfunction manifested as impaired ATP production, increased oxidative stress, disrupted membrane potential, and altered dynamics plays an active role in supporting tumor development and progression. The dual role of mitochondria in both energy supply and cell death regulation positions them as critical nodes in the balance between cancer cell survival and demise. In conclusion, mitochondrial function and dysfunction are intricately involved in the metabolic reprogramming of cancer cells. While cancer cells often upregulate glycolysis, many retain or even enhance mitochondrial oxidative phosphorylation to meet their high energy and biosynthetic demands. Mitochondrial abnormalities including altered dynamics, reduced ATP production, and retrograde signaling contribute to tumorigenesis and offer potential targets for therapy. A deeper understanding of these processes will be vital for developing novel and effective cancer treatments.

## 3. Alterations in ATP Production in Cancer Cells

A hallmark of cellular life is the ability to efficiently convert nutrients into usable energy. In eukaryotic cells, this is primarily accomplished through adenosine triphosphate (ATP) production via two major pathways: mitochondrial oxidative phosphorylation (OXPHOS) and cytosolic glycolysis. In normal cells, ATP production is predominantly driven by mitochondrial OXPHOS under normoxic conditions. However, cancer cells exhibit altered energy metabolism that often includes elevated glycolysis even in the presence of oxygen, a phenomenon widely known as the Warburg effect. Yet, modern research has challenged the oversimplification of cancer as exclusively glycolytic, revealing a more nuanced metabolic landscape where ATP production is flexible, dynamic, and heavily context-dependent [11]. A foundational study explored the total ATP production dynamics in MCF7 breast cancer cells under different nutrient conditions using Seahorse XF real-time analysis [12]. They demonstrated that MCF7 cells could maintain significantly elevated ATP production when provided with a mixture of glucose, glutamine, and pyruvate, as opposed to single substrates. Interestingly, this enhancement in ATP production was mainly attributed to increased oxidative ATP production, not glycolytic flux, challenging the classical Warburg-centric perspective. The findings suggest that cancer cells retain a robust capacity for mitochondrial ATP synthesis and actively use it to meet high biosynthetic and proliferative demands [12]. In stark contrast, non-cancerous myoblasts (C2C12 cells) exhibited no change in ATP production rate under varying substrate conditions, implying that cancer cells possess a unique metabolic plasticity that normal cells lack. This metabolic flexibility allows cancer cells to utilize multiple carbon sources to optimize ATP generation depending on availability. Importantly, the increase in oxidative ATP production occurred rapidly within minutes, suggesting that this was a functional response rather than one requiring transcriptional remodeling [12].

The relevance of ATP levels in the survival and adaptability of cancer cells was further underscored by a study that investigated ATP production in chemoresistant colon cancer cell lines (HT29-OxR and HCT116-OxR) [13]. These cells exhibited defective mitochondrial ATP production but compensated for this deficit by upregulating aerobic glycolysis. Paradoxically, even with impaired mitochondria, the chemoresistant cells displayed higher intracellular ATP levels than their drug-sensitive parental counterparts. This increase in ATP supported enhanced signaling via hypoxia-inducible factor 1-alpha (HIF-1α), contributing to drug resistance. The study coined the term “ATP debt” to describe the additional energy required by cancer cells to maintain survival signaling under genotoxic stress. Intriguingly, direct ATP delivery into cells was sufficient to induce drug resistance in previously sensitive cells, while ATP depletion via glycolytic inhibition resensitized resistant cells to chemotherapy [13]. These results highlight a critical distinction: although many cancer cells have dysfunctional mitochondria, they do not necessarily exhibit low ATP levels. Instead, they rewire metabolic pathways to maintain or even elevate ATP levels through enhanced glycolysis or auxiliary pathways. This capacity to uphold energy homeostasis under stress is likely a major factor behind cancer cell resilience and treatment resistance.

Another pathway contributing to ATP production in cancer cells—beyond glycolysis and classical OXPHOS—is the serine, one-carbon, and glycine (SOG) metabolism pathway [14]. This pathway branches off from glycolysis and shuttles carbon units through folate metabolism and glycine cleavage, ultimately generating ATP, NADPH, and purine nucleotides. Using metabolic flux modeling and tracer analysis, the authors showed that the SOG pathway is significantly upregulated in a subset of breast, prostate, colorectal, and lung cancers. Notably, the high expression of SOG pathway genes strongly correlated with increased proliferation and Myc oncogene activation, both indicators of aggressive tumor behavior [14]. The SOG pathway was predicted to produce up to four molecules of ATP per glucose molecule via its unique integration of glycolytic intermediates with mitochondrial one-carbon metabolism. The inhibition of this pathway using methotrexate led to a marked decrease in ATP levels and the activation of AMP-activated protein kinase (AMPK), confirming the pathway’s role in energy generation. These findings suggest that in some cancers, the SOG pathway acts as a third route for ATP production, complementing glycolysis and OXPHOS to ensure sufficient energy for growth and survival.

When comparing cancer cells to normal cells, a recurring theme is the flexibility of ATP production in tumors. While normal cells generally rely on OXPHOS under normoxic conditions and switch to glycolysis only under hypoxia, cancer cells often operate both pathways simultaneously or switch between them more readily. This adaptability allows them to maintain high ATP output even in hostile microenvironments where oxygen or nutrients may be limited [11,12]. In summary, cancer cells exhibit a dynamic and robust system for ATP production that surpasses that of normal cells in flexibility and capacity. While OXPHOS remains crucial in both cell types, cancer cells supplement mitochondrial ATP synthesis with enhanced glycolysis and alternative metabolic routes like the SOG pathway. Elevated intracellular ATP levels not only fuel biosynthesis and proliferation but also modulate survival signaling and therapy resistance. Understanding these mechanisms provides a strong rationale for the development of metabolic therapies aimed at disrupting ATP homeostasis in tumors.

## 4. Fuel Flexibility and Metabolic Reprogramming in Cancer: Utilization of Glucose, Glutamine, and Fatty Acids

Cancer cells undergo profound metabolic reprogramming to meet the demands of rapid proliferation, survival in harsh environments, and the evasion of immune responses. A major feature of this metabolic shift is fuel flexibility: the capacity to switch between different energy substrates such as glucose, glutamine, and fatty acids. This section explores how cancer cells utilize each of these fuels and how Seahorse XF Analyzer-based bioenergetic profiling has illuminated these adaptive strategies [15,16].

### 4.1. Glucose Metabolism and the Warburg Effect

Glucose is a primary substrate for energy generation in both normal and cancerous cells. In cancer, however, glucose metabolism is frequently rerouted toward aerobic glycolysis, a phenomenon known as the Warburg effect [15]. Despite the availability of oxygen, cancer cells convert glucose to lactate, sacrificing efficiency (less ATP per molecule of glucose) for speed and anabolic precursor generation. This metabolic phenotype supports the biosynthetic needs of rapidly proliferating cells. The Seahorse XF Analyzer has been instrumental in quantifying this glycolytic shift by measuring the extracellular acidification rate (ECAR), which correlates with lactate production and glycolytic flux [15]. In the Seahorse Mito Stress Test, glycolytic activity is often assessed following the inhibition of oxidative phosphorylation (OXPHOS) with drugs like oligomycin. In lupus-associated B and T cells, which show similarities to cancer metabolism, the inhibition of glycolysis using 2-deoxyglucose (2DG) significantly reduces ECAR, indicating high glycolytic dependency [15,16]. In addition to providing energy, glucose metabolism affects intracellular pH. A study published in *Cancer Research* revealed that metabolic reprogramming in cancer also serves to regulate proton production. Across multiple cancer types, altered metabolism leads to increased intracellular acidification, which compensates for chronic alkaline stress driven by inflammation and iron overload [17].

### 4.2. Glutamine as a Carbon and Nitrogen Source

When glucose availability is limited, cancer cells often rely on glutamine, an amino acid that feeds into the tricarboxylic acid (TCA) cycle via conversion to α-ketoglutarate. Glutamine serves as both a carbon donor for mitochondrial respiration and a nitrogen source for biosynthesis [16]. The enzyme glutaminase (GLS), which catalyzes the conversion of glutamine to glutamate, is critical for glutaminolysis. The inhibition of GLS using BPTES has been shown to reduce mitochondrial oxygen consumption rate (OCR), indicating the importance of glutamine in fueling OXPHOS [16]. Seahorse Fuel Flex assays, which track changes in OCR following the sequential addition of inhibitors like BPTES, enable researchers to quantify this dependency in real time [15]. In human corneal endothelial cells (cHCECs), a model for mitochondrial bioenergetics, mature differentiated cells demonstrated elevated OCR after glutamine supplementation, while immature or stressed cell populations did not show the same capacity for glutaminolysis. This finding highlights the link between mitochondrial function, cell differentiation, and glutamine utilization [18].

### 4.3. Fatty Acid Oxidation (FAO) as a Backup Fuel Source

Fatty acids are another key energy source, especially in nutrient-deprived environments. Fatty acid oxidation (FAO) supplies acetyl-CoA for the TCA cycle, supporting ATP production and redox homeostasis. In cancer, FAO contributes to metastatic potential and survival under oxidative stress [16]. Etomoxir, an inhibitor of carnitine palmitoyl transferase 1 (CPT1), is used in Seahorse assays to block FAO and assess its contribution to OCR. For instance, in glioblastoma tumor spheres, etomoxir reduced both OCR and invasive properties, illustrating the critical role of FAO in maintaining aggressive phenotypes [16]. The Seahorse Fuel Flex test also allows for the evaluation of spare respiratory capacity by observing OCR after FAO inhibition. This capacity reflects the cell’s ability to respond to increased energy demand and has been linked to tumor resilience and therapy resistance [15].

### 4.4. Metabolic Redundancy and Adaptation

Cancer cells often exhibit metabolic redundancy, enabling them to switch between glucose, glutamine, and FAO pathways depending on nutrient availability or therapy pressure. This adaptability is a hallmark of treatment resistance and tumor progression. Seahorse XF Fuel Flex assays have revealed that some cancer cells display primary dependence on glucose, while others show a compensatory increase in glutamine or fatty acid utilization when one pathway is inhibited [15]. Such metabolic switching is influenced by oncogenes like MYC and transcription factors like HIF-1α, which regulate genes involved in glycolysis and glutaminolysis [16,17]. Furthermore, studies in cHCECs have shown that cells with higher “effector cell” ratios—indicators of maturity and functionality—prefer OXPHOS and utilize glutamine and fatty acids more efficiently, while stressed or undifferentiated cells default to glycolysis [18].

### 4.5. Oncometabolites and Mitochondrial Metabolic Reprogramming

Mutations in isocitrate dehydrogenase (IDH), particularly in IDH1 and IDH2, have emerged as critical molecular events in various cancers, most notably gliomas, acute myeloid leukemia (AML), and chondrosarcomas [19]. These mutations reprogram mitochondrial metabolism and redox homeostasis, introducing unique vulnerabilities that may be exploited for therapeutic gain. In their wild-type form, IDH1 and IDH2 catalyze the oxidative decarboxylation of isocitrate to α-ketoglutarate (α-KG), simultaneously producing NADPH, a key reductant essential for detoxifying reactive oxygen species (ROS) and maintaining cellular redox balance. However, cancer-associated mutations, most commonly IDH1-R132H or IDH2-R140Q/R172K, confer a neomorphic enzymatic function. Instead of producing α-KG, the mutant enzymes convert α-KG and NADPH into the oncometabolite D-2-hydroxyglutarate (D-2-HG), with the concomitant generation of NADP^+^. This seemingly subtle alteration triggers profound metabolic and epigenetic consequences [20].

The accumulation of D-2-HG acts as a competitive inhibitor of α-KG–dependent dioxygenases, including the TET family of DNA demethylases and Jumonji-domain histone demethylases. As a result, IDH-mutant tumors exhibit global hypermethylation, altering transcriptional programs associated with metabolism, proliferation, and cell fate. In gliomas, for example, this epigenetic reprogramming reinforces a progenitor-like, stem-cell state, sustaining cancer stem cell (CSC) populations that are resistant to differentiation cues and standard therapies. Moreover, D-2-HG impairs the function of succinate dehydrogenase (SDH), resulting in succinyl-CoA accumulation and mitochondrial protein hypersuccinylation. This disrupts the mitochondrial membrane potential, stabilizes anti-apoptotic proteins such as BCL-2, and further protects cells from programmed cell death [21,22].

Beyond epigenetics, IDH mutations exert direct effects on mitochondrial function and bioenergetics. The depletion of NADPH undermines cellular antioxidant capacity, rendering cells more vulnerable to ROS accumulation, particularly under oxidative stress induced by radiation or chemotherapy. This redox vulnerability paradoxically correlates with the improved prognosis observed in patients with IDH1-mutant gliomas, likely due to their increased susceptibility to therapy-induced oxidative damage. However, this same vulnerability also raises therapeutic challenges; for instance, using IDH1 inhibitors during radiotherapy may restore NADPH pools, unintentionally shielding cancer cells from ROS-based treatments [23].

Metabolically, IDH1/2 mutations initiate a shift from the classic Warburg effect characterized by glycolysis dependence despite oxygen availability to increased reliance on oxidative phosphorylation (OXPHOS). In gliomas harboring IDH1-R132H mutations, cells exhibit increased mitochondrial mass, enhanced oxygen consumption, and preferential utilization of substrates like glutamate and pyruvate for ATP production. This rewiring is especially evident in glioma stem cells (GSCs), which display an OXPHOS-dominant phenotype, reinforcing their therapy-resistant and self-renewing properties. Importantly, CD133—a marker of stemness—has been found to be upregulated in high-grade IDH1-mutant gliomas, further linking mutant IDH metabolism to cancer stem cell maintenance [20]. Even though direct studies on IDH1-mutant GSCs are limited, accumulating evidence suggests that both differentiated tumor cells and GSCs share this OXPHOS-centric metabolic profile, providing a common vulnerability.

This phenomenon is not exclusive to gliomas. In AML, IDH1-mutant preleukemic hematopoietic stem cells (pHSCs) display impaired differentiation and a transcriptional signature distinct from both healthy HSCs and fully transformed leukemic stem cells (LSCs). A study demonstrated that these mutant pHSCs heavily depend on mitochondrial respiration and are selectively sensitive to OXPHOS inhibitors, such as IACS-010759, a complex I inhibitor [19]. Interestingly, these cells were not eradicated by the IDH1-specific inhibitor ivosidenib, underscoring that metabolic dependencies, rather than mutant enzyme activity per se, may be the more actionable target in early-stage disease. Moreover, this suggests a window of opportunity to eliminate mutant precursors before full transformation, potentially preventing disease progression or relapse [19].

At a biochemical level, the production of 2-HG disrupts not only signaling pathways and epigenetic regulation but also the structural and functional integrity of mitochondria. The resulting redox imbalance from NADPH depletion impairs the detoxification of ROS and leads to mitochondrial dysfunction. However, instead of succumbing to this stress, IDH-mutant cells often adapt by enhancing mitochondrial respiration efficiency. In mutant IDH2-expressing cancer models, such cells demonstrate increased ATP production through OXPHOS and suppressed glycolytic flux, particularly under hypoxic conditions. This adaptation facilitates tumor survival under stress and supports cellular energy demands without triggering a bioenergetic crisis. Yet, it also introduces metabolic rigidity, an Achilles’ heel that can be therapeutically exploited [24].

Therapeutic strategies are now being designed to target these metabolic liabilities. Metformin and phenformin, for instance, inhibit complex I of the electron transport chain and have demonstrated selective toxicity in OXPHOS-dependent IDH1-mutant tumor cells. Additional metabolic inhibitors—such as CB-839 and its prodrug JHU083—target glutamine metabolism, which is critical for α-KG replenishment and mitochondrial function in IDH1-mutant cells. Chloroquine and EGCG inhibit glutamate dehydrogenase, further disrupting the anaplerotic flux that sustains 2-HG production. These agents act synergistically to compromise energy generation and redox balance, selectively inducing apoptosis in IDH1-mutant tumors while sparing normal cells [23].

Yet, due to the essential role of mitochondria in healthy tissues, such therapies must be precisely delivered to minimize off-target toxicity. Novel drug delivery systems, including mitochondria-targeted prodrugs and nanocarriers, are being explored to enhance specificity and therapeutic index. Furthermore, researchers are investigating whether combining OXPHOS inhibitors with IDH-targeted or epigenetic therapies could provide dual-pronged attacks on both the metabolic and transcriptional aberrations driven by IDH mutations.

## 5. Molecular Dysregulation of Mitochondrial Dynamics in Cancer

### 5.1. Alterations in Mitochondrial Dynamics in Cancer

Mitochondria are highly dynamic organelles, constantly undergoing fission and fusion to maintain proper morphology, distribution, and function. This process, known as mitochondrial dynamics, plays a critical role in maintaining cellular energy homeostasis, bioenergetics, signaling, and apoptosis. In healthy cells, a balanced cycle of fission and fusion ensures mitochondrial quality control and adapts mitochondrial networks to metabolic needs [25]. However, in cancer cells, these dynamics are profoundly altered, contributing to tumorigenesis, progression, metastasis, and resistance to therapies. The disruption of this balance can reshape mitochondrial structure and localization, leading to metabolic reprogramming and altered cellular behavior [26]. The two major mitochondrial dynamics processes—fusion and fission—are mediated by specific sets of proteins: mitofusin 1 and 2 (MFN1, MFN2) and optic atrophy protein 1 (OPA1) regulate fusion, while fission is primarily controlled by dynamin-related protein 1 (Drp1), mitochondrial fission 1 protein (FIS1), mitochondrial fission factor (MFF), and other regulators such as mitochondrial fission process protein 1 (MTFP1). Alterations in the expression or activity of these proteins are commonly observed in a variety of cancer types and are increasingly recognized as critical modulators of malignant behavior [25,26,27].

Genomic analyses show that genes encoding mitochondrial dynamics regulators are recurrently amplified in multiple cancers, including breast, lung, pancreatic, and head and neck cancers. Specifically, amplifications of OPA1, MFN1, and DNM1L (encoding Drp1) were found in more than 5% of high-grade serous ovarian cancers (HGSOC), breast carcinomas, and lung adenocarcinomas [25]. These genetic alterations can modify the mitochondrial morphology towards either excessive fragmentation or hyperfusion, influencing tumor behavior.

Metastatic breast cancer cells were found to exhibit increased mitochondrial fragmentation compared to non-metastatic cells. This was due to elevated Drp1 levels and decreased Mfn1 expression. Drp1 not only mediated mitochondrial fission but also facilitated the redistribution of mitochondria to the lamellipodial regions of cancer cells, where energy is most needed for motility. Knockdown of Drp1 or overexpression of Mfn1 resulted in elongated mitochondria and significantly reduced cell migration and invasion in vitro. Conversely, silencing Mfn1 and Mfn2 promoted fragmentation and enhanced metastatic behavior [26]. Mutant oncogenes also modulate mitochondrial dynamics. For instance, in pancreatic ductal adenocarcinoma (PDAC), mutant KRAS signaling promotes mitochondrial fragmentation via enhanced Drp1 activity, which is essential for KRAS-driven tumor growth [25]. These changes in mitochondrial shape are not merely structural but play a central role in regulating apoptosis, cell metabolism, and adaptation to stress, which collectively promote malignancy. These findings emphasize the connection between mitochondrial dynamics and cancer cell motility. Mitochondrial fission appears necessary for redistributing mitochondria to cellular protrusions, such as lamellipodia, where energy-demanding activities like actin remodeling and focal adhesion turnover occur. Moreover, the pharmacological inhibition of Drp1 using Mdivi-1 suppressed migration, offering a potential therapeutic angle.

In a separate investigation into oncocytic thyroid tumors, abnormal mitochondrial accumulation and morphology were observed as hallmarks of these cancers. Oncocytic tumors are known for their high mitochondrial content, and this study demonstrated the upregulation of both fusion (Mfn1, Mfn2, Opa1) and fission (Drp1, Fis1) proteins. Notably, Drp1 overexpression was specifically associated with malignant transformation and increased migration potential in thyroid tumor cells [27]. The genetic and pharmacologic inhibition of Drp1 reduced the migratory ability of oncocytic thyroid cancer cells, suggesting that Drp1 plays a role beyond mitochondrial morphology—this may directly impact metastatic capacity. The study reinforced the concept that imbalanced mitochondrial dynamics favoring fission is linked with increased tumor aggressiveness. Interestingly, these alterations occurred despite similar levels of mitochondrial content (as normalized by SDHA staining), indicating that it is not mitochondrial mass but dynamics and structure that determine cancer cell behavior [27].

These results align with broader findings across cancers where fragmented mitochondrial networks are linked to malignancy. For instance, mitochondrial fragmentation has also been observed in aggressive renal cell carcinomas and glioblastomas, supporting the idea that enhanced fission is a conserved feature of metastatic tumors.

### 5.2. p53, Drp1, and Metastatic Potential

The tumor suppressor p53, widely known for its roles in DNA repair and apoptosis, has been shown to regulate mitochondrial dynamics. p53 suppresses the expression of MTFP1, a protein that promotes mitochondrial fission by enhancing Drp1 phosphorylation at Ser616. The loss of p53 leads to exaggerated mitochondrial fragmentation, which correlates with aggressive cancer phenotypes, epithelial-to-mesenchymal transition (EMT), and enhanced metastatic capacity [28]. Mechanistically, the absence of p53 activates mTORC1 signaling, upregulating MTFP1, which then facilitates Drp1-driven mitochondrial fission. This pathway, termed the mTORC1/MTFP1/Drp1 axis, culminates in increased ERK1/2 activation, MMP9 expression, and cell migration. Interfering with this axis, either genetically or pharmacologically, inhibits these metastatic phenotypes, suggesting therapeutic potential. Furthermore, wild-type p53 enhances mitochondrial fusion and elongation, thereby suppressing invasive cell migration. This highlights the dual role of p53 as both a genomic and mitochondrial gatekeeper in tumor progression [28].

### 5.3. Dynamic Interplay and Context-Specific Effects

Despite the overarching trends, the biological outcomes of mitochondrial fission or fusion vary based on cancer type and cellular context. For example, promoting mitochondrial fusion using P-Mito compounds was shown to suppress pancreatic tumor growth and induce chromosomal instability due to persistent hyperfusion [29]. On the other hand, inhibiting fission via Drp1 downregulation diminished tumor proliferation and migration, pointing to its potential as a therapeutic target. In estrogen receptor-positive breast cancers, fusion proteins OPA1 and MFN2 were found to be downregulated in comparison to normal mammary epithelial cells, suggesting a predisposition to a fragmented mitochondrial state. This fragmented state is often associated with poor prognosis and drug resistance [29]. However, the complexity arises from paradoxical findings such as mitochondrial fission reducing metastasis in certain cancers like triple-negative breast cancer (TNBC). This context-dependent behavior suggests that therapeutic strategies must consider tumor subtype and mitochondrial status before modulation [28].

### 5.4. Mitochondrial Trafficking and Cortical Localization

Mitochondrial positioning within the cell is another emerging theme in cancer dynamics. A study by Caino et al. (2016) revealed that metastatic tumor cells co-opt a neuronal mitochondrial trafficking network, typically active in neurons, to redistribute mitochondria to the cortical cytoskeleton. This network includes syntaphilin (SNPH), kinesin motor proteins (like KIF5B), and GTPases such as Miro1 and Miro2 [30]. The authors identified SNPH as a critical brake on mitochondrial motility. In tumors, SNPH is frequently downregulated, allowing mitochondria to move more rapidly and accumulate in cell protrusions. This redistribution fuels localized ATP production, facilitating chemotaxis, invasion, and metastasis. Silencing SNPH increased both mitochondrial fission rates and fusion events, suggesting that hyperactive mitochondrial dynamics contribute to metastatic capacity. Importantly, restoring SNPH expression suppressed invasion and cell migration in multiple cancer models [30]. Their findings suggest that mitochondrial fission in cancer is not just a structural change but part of a coordinated system that facilitates energy delivery to the leading edge of migrating tumor cells. This spatial regulation of mitochondrial function may be especially important in cells undergoing epithelial-to-mesenchymal transition (EMT), which requires dynamic cytoskeletal remodeling and significant energy expenditure.

Mitochondrial dynamics represent a vital layer of regulation in cancer biology, linking bioenergetics with cell structure, signaling, and motility. The imbalance between fission and fusion—particularly through the upregulation of Drp1 and downregulation of MFN1/MFN2—facilitates mitochondrial fragmentation and subcellular relocalization, a feature strongly associated with metastatic potential [29,30]. In several cancer types, including breast and thyroid tumors, this shift in dynamics correlates with increased invasion and poorer prognosis. Additionally, cancer cells exploit neuronal-like mitochondrial trafficking mechanisms to position mitochondria at sites of high energy demand, such as the cortical cytoskeleton, further enhancing their invasive capabilities and highlighting the intricate nature of mitochondrial behavior in malignancy [26,29].

These alterations in mitochondrial dynamics reflect a finely tuned imbalance regulated by core proteins including Drp1, OPA1, MFN1/MFN2, and MTFP1 [25]. The resulting structural and functional changes influence cancer cell survival, proliferation, and migration, often involving critical oncogenic pathways such as p53 and mTORC1. Importantly, this dysregulation introduces specific therapeutic vulnerabilities that can be targeted using pharmacological agents designed to modulate mitochondrial morphology and function. Inhibiting excessive fission, restoring fusion balance, or blocking mitochondrial trafficking may all disrupt the energetic and structural support systems that drive tumor aggressiveness [28].

Given the diversity in mitochondrial behavior across cancer types, future research should prioritize the identification of biomarkers that reflect mitochondrial dynamic states and predict therapeutic responses. Integrating mitochondrial-targeted therapies with apoptosis-inducing agents may offer synergistic benefits, particularly in treatment-resistant cancers characterized by p53 mutations or aberrant KRAS signaling. Understanding and targeting mitochondrial dynamics not only enhances our knowledge of cancer progression but also opens the door to novel anti-metastatic strategies that disrupt the cellular energy architecture underpinning tumor invasion and dissemination.

## 6. Mitochondrial Morphology in Cancer Cells

Mitochondria, commonly known as the cell’s powerhouse, display a dynamic and flexible morphology that mirrors both physiological and pathological conditions. Their shape results from the finely tuned balance between mitochondrial fission and fusion, processes that are crucial for sustaining cellular energy production, redox balance, and cell survival [25]. In cancer cells, this delicate equilibrium is frequently disrupted, leading to profound alterations in mitochondrial shape, distribution, and ultrastructure. These morphological changes are increasingly recognized not only as hallmarks of metabolic reprogramming and tumor progression but also as active drivers of drug resistance and adaptation to the tumor microenvironment. Influenced by factors such as oxidative stress, altered membrane potential, and oncogenic signaling, mitochondrial remodeling contributes significantly to the malignant phenotype and represents a critical feature of cancer cell biology [31,32,33].

For optimal bioenergetic function and cellular homeostasis, mitochondria in healthy eukaryotic cells maintain a dynamic and tightly controlled morphology. Usually found in metabolically active cells, mitochondria are long, tubular organelles that form an interconnected network. This shape is dynamic; rather, the size, number, and distribution of mitochondria are constantly balanced by fusion and fission events, which are collectively referred to as mitochondrial dynamics (Figure 1). Complementation between partially damaged organelles is facilitated by mitochondrial fusion, which permits the mixing of mitochondrial contents such as proteins, metabolites, and mitochondrial DNA (mtDNA). On the other hand, the ability to isolate damaged mitochondrial segments for removal through mitophagy relies on fission. This process facilitates quality control and the redistribution of mitochondria during cell division or in response to changes in metabolic demand. Together, these mechanisms preserve mitochondrial flexibility and integrity across various physiological conditions [30]. Healthy mitochondria possess a double-membrane structure, with an inner mitochondrial membrane (IMM) that folds inward to form cristae and an outer mitochondrial membrane (OMM) that encloses the organelle (Figure 1). By containing vital elements of the electron transport chain (ETC) and ATP synthase required for oxidative phosphorylation, these cristae greatly expand the membrane surface area. The maintenance of cristae architecture is crucial for controlling reactive oxygen species (ROS), regulating calcium homeostasis, and ensuring efficient ATP synthesis. Healthy mitochondria display intact membranes and well-defined, closely spaced cristae under electron microscopy, indicating robust bioenergetic capacity [32]. Mitochondrial dysfunction is frequently linked to changes in this typical morphology, such as excessive fragmentation, swelling, or loss of cristae structure. These alterations have been noted in diseases such as cancer, neurodegeneration, and infertility. For example, aberrant fission that results in mitochondrial fragmentation has been connected to the increased production of ROS and impaired oxidative phosphorylation, both of which contribute to the development of disease and cellular damage [31,32] Additionally, because structural disruptions are directly linked to metabolic abnormalities, sophisticated imaging studies have emphasized the significance of preserving mitochondrial morphology in live-cell contexts [34].

Therefore, mitochondria’s distinctive shape—a balance of linked tubules with preserved cristae—reflects their functional state. A crucial factor in comprehending both normal cell function and the etiology of different disorders is the proper regulation of mitochondrial shape, which is not only structural but also closely related to mitochondrial and cellular physiology.

### 6.1. Morphological Shifts During Tumor Progression

One of the most compelling illustrations of morphological transformation occurs in ovarian cancer progression. In a study utilizing the mouse ovarian surface epithelial (MOSE) model, mitochondrial morphology transitioned from a filamentous network in benign cells (MOSE-E) to fragmented and enlarged spherical structures in highly aggressive tumor-initiating cells (MOSE-LTICv) (Figure 2) [33]. These changes were attributed to an imbalance in the expression of mitochondrial dynamics proteins: mitofusin 1 (MFN1) and OPA1 (fusion mediators) were downregulated, while fission-related proteins such as DRP1 and FIS1 were upregulated (Figure 2). The transformation coincided with adaptation to hypoxic environments, where fragmented mitochondria facilitated enhanced autophagy and ROS regulation, supporting cell survival under stress [33].

Another study has shown that mitochondrial morphology varies greatly among different cancer types and can be used to characterize the bioenergetic and therapeutic response profiles of tumor cells. In a study, mitochondrial phenotypes in both cancer cell lines and patient-derived xenografts were categorized into three primary morphological classes: punctate (fragmented), intermediate, and filamentous (elongated) [36]. Each phenotype correlated with specific metabolic features. For example, cells displaying a predominantly punctate mitochondrial network tended to have lower oxidative phosphorylation (OXPHOS) activity and a higher reliance on glycolysis, indicating a metabolic reprogramming consistent with the Warburg effect [36].

Their imaging-based computational pipeline provided a robust quantification of mitochondrial morphology and showed that morphological phenotypes remain consistent over time within a cell population, even across different cancer origins. Furthermore, treatments like ABT-263 and cisplatin shifted mitochondrial networks toward a more punctate state in drug-sensitive cancers, implying that mitochondrial morphology is sensitive to therapeutic intervention and may serve as an early biomarker of drug efficacy [35]. Also, a study demonstrated that mitochondrial fragmentation was localized around the nucleus in malignant cells, with decreased branching, shorter mean branch length, and increased circularity. This contrasted starkly with the elongated, well-distributed mitochondrial networks of benign cells. Interestingly, while hypoxia had a modest effect on mitochondrial morphology, the degree of malignancy was the primary determinant of mitochondrial form [34].

### 6.2. Ultrastructural Deformities and Functional Implications

Transmission electron microscopy (TEM) studies provide further insight into mitochondrial architecture at the nanoscale. In retinoblastoma specimens, poorly differentiated tumors exhibited mitochondria with markedly swollen appearances, disorganized or dissolved cristae (cristolysis) (Figure 3), and even mitochondrial ghost structures—remnants without discernible internal membranes [36]. These changes were more pronounced in tumors exhibiting necrosis or high invasiveness, implying a correlation between structural degradation and tumor aggressiveness. Additionally, retinoblastoma cells demonstrated signs of early apoptosis and necroptosis alongside mitochondrial deformation. Swollen mitochondria with dense matrices, disrupted outer membranes, and cristae loss reflect severe dysfunction in bioenergetic output and protein import systems. These abnormalities likely arise from impaired oxidative phosphorylation (OXPHOS), consistent with a shift toward glycolytic metabolism, characteristic of the Warburg phenotype (Figure 3) [37].

### 6.3. Quantitative Assessment Using Fractal Analysis

To complement qualitative classification, a study utilized fractal geometry analysis to objectively assess mitochondrial morphology in malignant mesothelioma (MM) [38]. Using fractal dimension and lacunarity, they quantified the complexity and texture of mitochondrial networks in different MM cell lines and compared them with control mesothelial cells. Control MeT-5A cells displayed elongated and highly interconnected mitochondria, characterized by high fractal dimension and low lacunarity, representing a well-organized mitochondrial network. In contrast, mesothelioma cells such as H513 and H2596 showed lower fractal dimension and higher lacunarity, indicative of more fragmented, less space-filling networks [36]. Interestingly, these morphological changes also correlated with therapeutic response. MM cells with lower fractal dimensions and more fragmented mitochondria were more sensitive to metformin, an OXPHOS inhibitor, and mdivi-1, a Drp1 inhibitor targeting mitochondrial fission. Meanwhile, mitochondrial morphology was a better predictor of drug sensitivity than OCR (oxygen consumption rate) or ECAR (extracellular acidification rate), commonly used metabolic readouts. This highlights that structural assessments of mitochondria may have more predictive value in certain contexts than metabolic assays alone. A more granular look into mitochondrial morphology was presented in a study conducted a comprehensive electron microscopy study of skeletal muscle mitochondria, comparing intermyofibrillar (IMF) and subsarcolemmal (SS) populations. Although their study focused on muscle, the methods and descriptors—circularity, aspect ratio, form factor, and membrane contact points—are highly applicable to cancer cell analysis. Picard et al. found that mitochondrial morphology is not static; it exists on a spectrum influenced by metabolic activity, with more elongated and interconnected mitochondria observed in metabolically active regions. The presence of electron-dense contact points between adjacent mitochondria (putative fusion/fission sites) suggests dynamic interactions even in post-mitotic tissues. In cancer, these metrics often shift toward increased circularity and reduced aspect ratio, indicating fragmentation and functional compartmentalization. In the context of cancer, such morphometric tools offer a non-invasive approach to assess mitochondrial health and predict treatment responses. For instance, higher circularity and reduced branching may be predictive of cells reliant on glycolysis and prone to oxidative stress [39].

### 6.4. Mitochondrial Morphology and Drug Resistance

The relationship between mitochondrial morphology and chemoresistance is particularly evident in pancreatic ductal adenocarcinoma (PDAC). In a study by Dash et al. (2024), PDAC cells deficient in deoxycytidine kinase (DCK), a key enzyme involved in activating gemcitabine, exhibited remarkable upregulation of mitochondrial genes associated with OXPHOS [40]. These DCK-knockout (KO) cells had higher mitochondrial respiration and ATP production, correlating with enhanced resistance to gemcitabine. Notably, electron microscopy revealed abnormal mitochondrial morphology in DCK KO cells, providing strong evidence for the connection between morphological changes and metabolic re-programming [40]. Mitochondria in DCK-deficient cells were described as more compact and structurally altered, and these changes were accompanied by reduced ROS levels, the upregulation of antioxidant genes (SOD1, SOD2), and increased anti-apoptotic BCL2 expression. The morphological alterations were functionally significant as targeting these mitochondria with complex I inhibitors (e.g., IACS-010759) or BCL2 inhibitors (venetoclax) sensitized the resistant PDAC cells to treatment. Therefore, mitochondrial shape not only reflects bioenergetic adaptations but also has the potential to influence therapeutic vulnerabilities. Mitochondrial morphology also holds promise as a diagnostic biomarker. In retinoblastoma, poorly differentiated tumors consistently displayed fewer and less structured mitochondria compared to their well-differentiated counterparts [36]. These morphological traits could potentially be used to stratify patients by prognosis or guide therapy intensity. From a therapeutic standpoint, interventions targeting mitochondrial dynamics or morphology-modifying pathways (e.g., fission inhibitors like mDivi-1 or fusion promoters) could restore bioenergetic balance and trigger apoptosis. Furthermore, drugs targeting OXPHOS complexes may preferentially affect cancer cells with disrupted mitochondrial architecture and function.

## 7. Modulation of Mitochondria-Associated Membranes and Calcium Signaling in Cancer Cells

Mitochondria and the endoplasmic reticulum (ER) maintain a highly coordinated relationship through specialized subcellular structures known as mitochondria-associated membranes (MAMs) [41]. These contact sites, which span a narrow 10–25 nm distance without membrane fusion, serve as critical hubs for calcium (Ca^2+^) transfer, lipid metabolism, reactive oxygen species (ROS) signaling, and mitochondrial bioenergetics. Over the last decade, growing evidence has highlighted the pivotal role of MAMs in cancer biology, revealing how their structural integrity and functional modulation govern key processes such as metabolic reprogramming, apoptosis resistance, proliferation, and metastatic behavior [42].

The ER functions as the primary intracellular Ca^2+^ reservoir. Upon stimulation, Ca^2+^ is released through inositol 1,4,5-trisphosphate receptors (IP3Rs), particularly the IP3R3 isoform enriched at MAMs [43]. These Ca^2+^ ions are then transferred to mitochondria via the voltage-dependent anion channel (VDAC) on the outer mitochondrial membrane and the mitochondrial calcium uniporter (MCU) on the inner membrane [44]. This tightly regulated transfer ensures localized Ca^2+^ elevations at the MAM interface, which, in turn, activate mitochondrial dehydrogenases, promote oxidative phosphorylation (OXPHOS), and support the synthesis of ATP to meet cellular energy demands. However, sustained or excessive mitochondrial Ca^2+^ uptake can trigger mitochondrial dysfunction, permeability transition pore (mPTP) opening, ROS overproduction, and apoptosis, a mechanism that is often disrupted in cancer cells to favor survival and progression [42,44].

In cancer, the architecture and functional output of MAMs are frequently reprogrammed. One striking example is triple-negative breast cancer (TNBC). One study demonstrated that these cells exhibit spontaneous, constitutive, IP3-linked Ca^2+^ oscillations [43]. These signals are directly sensed by mitochondria through MAMs and have been shown to regulate fatty acid metabolism and cell migration, implicating them in the aggressive phenotype of TNBC. Interestingly, genetic ablation of the MCU in TNBC cells significantly impaired mitochondrial Ca^2+^ uptake and reduced their invasive potential. However, this did not severely affect ATP-linked oxygen consumption or cellular ATP levels, suggesting a decoupling between mitochondrial Ca^2+^ signaling and ATP generation in these cells, indicating that Ca^2+^ dynamics at MAMs in cancer may be repurposed more for signaling than for bioenergetics [43].

The oncogenic modulation of Ca^2+^ signaling extends beyond metabolic regulation and also encompasses apoptotic control. A study provided compelling evidence that the tumor suppressor p53, long recognized for its nuclear transcriptional activities, also exerts non-transcriptional functions at the ER and MAMs [41]. Under stress conditions, wild-type p53 accumulates at MAMs, where it binds to and activates the sarco/ER Ca^2+^-ATPase (SERCA) pump. This interaction increases ER Ca^2+^ loading and enhances Ca^2+^ transfer to mitochondria, ultimately leading to mitochondrial Ca^2+^ overload, organelle fragmentation, and apoptosis. In contrast, cancer-associated p53 mutants fail to stimulate this Ca^2+^ flux and therefore diminish the apoptotic response, underscoring the significance of MAM-localized p53 in maintaining pro-death signaling in normal cells and how its dysfunction facilitates tumor survival [41].

Beyond p53, several other oncogenes and oncogenic regulators influence MAM composition and function. For instance, the Sigma-1 receptor (SigmaR1), a chaperone protein enriched at MAMs, is overexpressed in several cancers, including breast and colorectal malignancies. SigmaR1 has been shown to stabilize complexes between the Ca^2+^-activated K^+^ channel SK3 and the plasma membrane Ca^2+^ channel Orai1. These complexes facilitate constitutive and store-operated Ca^2+^ entry (CCE and SOCE), maintaining elevated cytosolic Ca^2+^ levels that promote cell migration and metastasis. Importantly, this Ca^2+^ influx does not trigger apoptosis, suggesting that SigmaR1 contributes to cancer cell survival by finely tuning Ca^2+^ signaling without crossing the threshold for mitochondrial collapse [45].

Elevated mitochondrial Ca^2+^ uptake via MCU is another common feature in various malignancies. In colorectal cancer (CRC), MCU expression is significantly increased, promoting mitochondrial biogenesis and enhancing tumor growth. A study reported that MCU overexpression inhibits the phosphorylation of transcription factor A, mitochondrial (TFAM) and stabilizing mitochondrial DNA transcription and replication. This leads to higher mitochondrial content, enhanced OXPHOS, and ROS production, which together activate NF-κB signaling, a pathway closely linked to cancer cell proliferation. These findings indicate that mitochondrial Ca^2+^ is not merely a buffering ion but acts as a signaling hub that reinforces oncogenic transcriptional programs [44].

The pathological elevation of mitochondrial Ca^2+^ has also been exploited therapeutically. A study engineered a nanoagent that synergistically triggers mitochondrial Ca^2+^ overload in cancer cells through the near-infrared (NIR)-induced release of Fe^2+^ and H^+^, inducing photoacidification and ROS-mediated damage [46]. This approach results in Ca^2+^ imbalance, mitochondrial dysfunction, and tumor cell death, highlighting a promising strategy to force cancer cells beyond their adaptive threshold and re-sensitize them to apoptosis [46].

Interestingly, not all tumors uniformly elevate mitochondrial Ca^2+^ flux. In fibrolamellar carcinoma (FLC), for example, MCU upregulation suppresses branched-chain amino acid degradation and the urea cycle by downregulating the transcription factor KLF15, facilitating anabolic metabolism over catabolism and further promoting tumor growth [47]. This suggests that the oncogenic role of mitochondrial Ca^2+^ signaling is context-specific, capable of either enhancing bioenergetics or repressing metabolic pathways depending on the cellular demands and oncogenic landscape.

Further complexity is introduced by the involvement of Bcl-2 family proteins in regulating MAM-mediated Ca^2+^ dynamics. Anti-apoptotic members such as Bcl-2 and Bcl-XL inhibit Ca^2+^ transfer from the ER to mitochondria by binding to IP3Rs and VDAC1. This suppression of mitochondrial Ca^2+^ accumulation serves as a protective mechanism against apoptosis, contributing to chemoresistance and tumor persistence [42].

Moreover, mitochondrial dysfunction itself can feed back into Ca^2+^ signaling to reinforce tumorigenesis. In cells lacking mitochondrial DNA (ρ0 cells), the absence of OXPHOS results in elevated cytosolic Ca^2+^ levels. These levels activate NF-κB signaling, which downregulates the expression of tumor suppressor p53, contributing to drug resistance and malignancy in colorectal cancer. Intriguingly, this process is reversible by Ca^2+^ chelation, further emphasizing the therapeutic relevance of Ca^2+^ signaling modulation in cancer [48].

## 8. Recent Advances in Mitochondria-Targeted Therapeutics

### 8.1. IACS-010759

IACS-010759 is a potent, selective inhibitor of mitochondrial complex I (NADH-ubiquinone oxidoreductase) that has emerged as a promising therapeutic agent targeting oxidative phosphorylation (OxPhos) in cancer. Unlike classical quinone-site inhibitors such as piericidin or acetogenins, IACS-010759 inhibits complex I through a unique binding mechanism. It directionally inhibits both forward and reverse electron transfer within the respiratory chain, with approximately 10-fold greater potency for the reverse reaction (IC_50_ ~ 41 nM) compared to the forward (IC_50_ ~ 460 nM). This atypical mechanism involves binding to the ND1 subunit within complex I, rather than the canonical quinone-access channel, as shown by photoaffinity labeling and resistance mutations in ND1 (e.g., Leu55Phe) [49].

Functionally, IACS-010759 induces bioenergetic collapse in tumors heavily reliant on OxPhos. In chronic lymphocytic leukemia (CLL), which shows elevated OxPhos activity, especially in poor prognosis subtypes, IACS-010759 substantially reduces oxygen consumption rate (OCR) and intracellular nucleotide pools. However, it triggers compensatory glycolysis, limiting cytotoxicity when used alone. Combining IACS-010759 with glycolytic inhibitors such as 2-deoxy-D-glucose (2-DG) enhances its pro-apoptotic effects, indicating the need for dual-pathway inhibition in metabolically flexible tumors [50]. In T-cell acute lymphoblastic leukemia (T-ALL), particularly in NOTCH1-mutated subtypes, IACS-010759 induces metabolic shutdown, glutaminolysis, and redox imbalance, contributing to tumor suppression. Its efficacy is enhanced when combined with L-asparaginase, which unmasks glutamine dependency, suggesting a synergistic therapeutic approach [51].

### 8.2. Metformin and Phenformin

Metformin, a first-line antidiabetic drug, has garnered considerable interest in oncology due to its ability to inhibit mitochondrial complex I and induce metabolic stress in cancer cells [52,53]. Although metformin alone shows limited efficacy as an anticancer monotherapy, its mechanisms of action and potential in combinatorial strategies have made it a subject of intense research [54,55]. The related biguanide phenformin, previously withdrawn from the market due to lactic acidosis risk, has been revisited as a more potent mitochondrial complex I inhibitor with superior anticancer activity under controlled conditions [56].

Mechanistically, metformin directly inhibits mitochondrial complex I (NADH: ubiquinone oxidoreductase), thereby reducing oxygen consumption, ATP production, and mitochondrial NAD^+^/NADH ratio. In a foundational study, metformin was shown to suppress oxidative phosphorylation in cancer cells in vitro and in vivo [52]. Its inhibitory effects on proliferation and respiration were fully abrogated in cells expressing the yeast NADH dehydrogenase NDI1, confirming that complex I inhibition is essential for metformin’s anticancer activity. Importantly, metformin reduced tumor growth in mouse xenografts only when complex I was active in the tumor cells [52]. At a cellular level, metformin’s inhibition of complex I leads to decreased aspartate synthesis, the disruption of nucleotide biosynthesis, and impaired cell proliferation. Another study demonstrated that metformin lowers cytosolic aspartate levels indirectly through complex I inhibition, impairing tumor cell viability, especially under low-oxygen conditions [53]. They also showed that supplementation with pyruvate or aspartate could partially rescue these effects, emphasizing the role of mitochondrial respiration in redox homeostasis and nucleotide synthesis [53]. Beyond bioenergetics, metformin also alters mitochondrial ultrastructure and calcium signaling. Another study found that metformin induces ER stress, leading to calcium release into the cytoplasm and subsequent mitochondrial uptake via the mitochondrial calcium uniporter (MCU) [56]. This results in mitochondrial swelling, cristae disorganization, and the inhibition of mitochondrial permeability transition pore (mPTP) opening, contributing to mitochondrial dysfunction without triggering strong apoptosis. These findings suggest that metformin may promote a pro-survival stress response, which could limit its cytotoxicity unless paired with additional metabolic or apoptotic stressors [56].

Phenformin, a more hydrophobic and membrane-permeable analog of metformin, inhibits complex I more potently and at significantly lower concentrations. A study conducted a comparative metabolomics analysis of metformin and phenformin in models of cellular transformation and breast cancer stem cells (CSCs) [54]. Both drugs significantly depleted TCA cycle intermediates and selectively reduced nucleotide triphosphates in CSCs, but phenformin achieved these effects at 30-fold lower concentrations. Importantly, while phenformin and metformin shared many metabolic impacts, phenformin showed enhanced efficacy in suppressing the transformation process and selectively targeting CSCs, which are often resistant to standard therapies [54]. Clinical use of phenformin, however, is restricted due to its risk of inducing lactic acidosis, especially in patients with impaired renal function. Nevertheless, phenformin has shown promising results in combination therapies. Veiga et al. (2018) investigated phenformin in hepatocellular carcinoma (HCC) and found that it induces mitochondrial fragmentation, a shift toward glycolysis, and sensitizes tumors to the dual inhibition of mTOR using allosteric (RAD001) and ATP-competitive (BEZ235) inhibitors. Strikingly, pretreatment with phenformin improved the efficacy of mTOR-targeted therapy and dramatically prolonged survival in orthotopic liver tumor models. This highlights phenformin’s potential as a metabolic sensitizer, particularly in tumors with dysregulated mTOR signaling [55].

Small-molecule inhibitors targeting mitochondria have emerged as promising anticancer strategies by exploiting tumor-specific metabolic and apoptotic vulnerabilities. One notable agent, IACS-010759, is a selective mitochondrial complex I inhibitor that disrupts oxidative phosphorylation (OxPhos), particularly effective in tumors with high mitochondrial dependency. It induces metabolic collapse and enhances apoptosis when combined with glycolytic or glutamine-targeting agents. Similarly, metformin and the more potent phenformin inhibit complex I, impairing ATP production and redox balance. While metformin’s anticancer effects are modest, phenformin shows greater efficacy, especially in targeting cancer stem cells and enhancing mTOR inhibitor responses. Beyond complex I inhibitors, other mitochondria-targeting agents include Elesclomol, which induces mitochondrial ROS; venetoclax (ABT-199), a Bcl-2 inhibitor that promotes mitochondrial outer membrane permeabilization; and MitoQ, a mitochondria-targeted antioxidant that modulates redox-sensitive cell death pathways. Agents targeting the mitochondrial permeability transition pore (mPTP) are also under investigation for their ability to control apoptosis in tumor cells. Collectively, these compounds illustrate the therapeutic potential of disrupting mitochondrial metabolism, redox homeostasis, and apoptotic signaling in cancer, particularly when used in rational combinations tailored to metabolic dependencies.

## 9. Conclusions

Mitochondria are recognized as crucial regulators in cancer biology, controlling a wide range of cellular processes including metabolism, redox balance, apoptosis, and stress adaptation. They are no longer considered passive organelles solely responsible for ATP production. This review emphasizes the structural reprogramming and functional plasticity of mitochondria, which contribute to the malignant phenotype across various tumor types. While the Warburg effect—suggesting that cancer cells prefer glycolysis even in the presence of oxygen—has long been a focal point, substantial evidence now shows that mitochondrial oxidative phosphorylation (OXPHOS) remains active and essential in many cancers. Rather than simply shutting down mitochondrial respiration, cancer cells reprogram it to support growth, resist apoptosis, and survive in hypoxic or nutrient-deprived environments. In fact, in tumors with high metabolic demands, ATP synthesis via mitochondrial respiration often surpasses that of glycolysis. Additionally, the role of mitochondria in tumor energy metabolism is further extended by other bioenergetic pathways, such as fatty acid oxidation and the serine-one-carbon-glycine (SOG) pathway.

One of the most striking hallmarks of cancer cell mitochondria is metabolic flexibility. This enables tumor cells to switch between using fatty acid oxidation, glucose, and glutamine as fuel sources based on the surrounding environment. Such flexibility supports biosynthesis and redox homeostasis in addition to energy production. This metabolic reprogramming is closely related to mitochondrial respiration rates, substrate usage patterns, and even drug responses, underscoring its diagnostic and therapeutic significance. This review highlights the importance of mitochondrial dynamics—the ongoing cycle of fission and fusion—in cancer development and metastasis, in addition to its role in metabolism. Mitochondrial fragmentation, compromised OXPHOS, and an increased propensity for migration and invasion result from disturbances in this balance, especially through overactive Drp1-mediated fission and the loss of fusion regulators such as MFN1, MFN2, and OPA1. Since mitochondria are redistributed to the leading edge of migrating cells, providing localized ATP for motility, these alterations not only reflect tumor aggressiveness but also actively contribute to the potential of metastasis.

Crucially, genetic changes in regulators of mitochondrial dynamics, like the downregulation of p53, which typically inhibits fission, and amplifications of DNM1L (Drp1), further entrench cancer cells in a state that is resistant to apoptosis and pro-metastatic. Since not all tumors exhibit the same level of mitochondrial remodeling, context-specific assessment is crucial when thinking about mitochondrial-targeted treatments. In certain situations, suppressing fission blunts invasion and improves the therapeutic response, whereas encouraging fusion sometimes prevents proliferation.

From filamentous to punctate forms, mitochondrial morphology provides a real-time readout of bioenergetic states and the course of disease. Fractal analysis of mitochondrial networks and high-resolution imaging have shown that elongated, interconnected networks of mitochondria are usually found in benign or well-differentiated cells, whereas fragmented, spherical mitochondria are common in aggressive and drug-resistant tumors. In addition to serving as markers of the severity of the disease, these morphological characteristics may also forecast how well a treatment will work. Research conducted on retinoblastoma and pancreatic models has demonstrated a correlation between morphological disruption and ROS levels, OXPHOS activity, and therapeutic outcomes, thereby confirming mitochondria as both therapeutic targets and biomarkers.

Given these diverse functions, cancer cells’ mitochondria offer a special therapeutic opportunity. It may be possible to specifically reduce tumor viability while preserving normal cells by targeting mitochondrial ATP synthesis, changing dynamics through Drp1 or fusion protein modulation, or interfering with mitochondrial trafficking and positioning. In tumors with a high degree of mitochondrial dependency, combined approaches that combine metabolic inhibitors with pro-apoptotic agents or oxidative stress inducers may be especially successful.

In summary, mitochondria in cancer cells are not just malfunctioning metabolic remnants; rather, they are morphologically modified, functionally optimized, and strategically rewired to promote malignancy. We can discover new pathways for precise, mitochondria-targeted cancer therapies by further deconstructing their structural, metabolic, and regulatory complexity. This will ultimately improve prognosis and overcome treatment resistance.

## Figures and Tables

**Figure 1 ijms-26-06750-f001:**
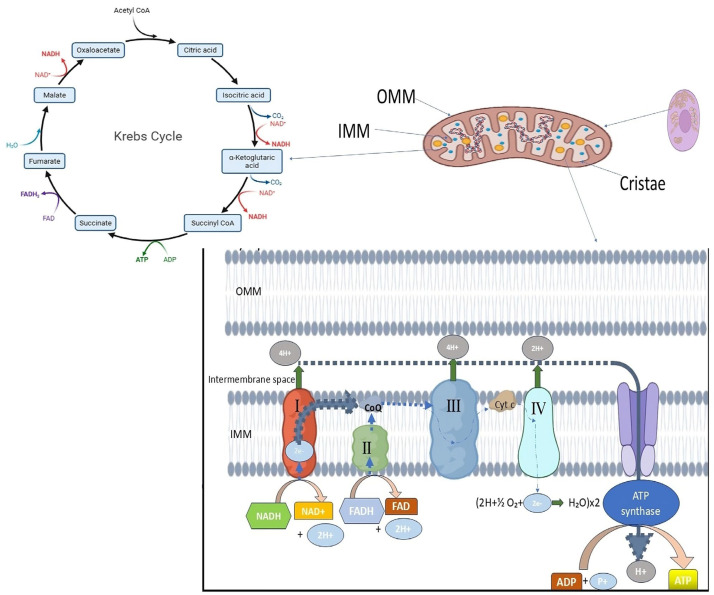
Normal mitochondrial morphology in a high-metabolic-activity cell. The figure illustrates mitochondria with a typical shape and size featuring a double-membrane structure composed of the outer mitochondrial membrane (OMM) and the inner mitochondrial membrane (IMM). The IMM forms inward folds known as cristae, where the electron transport chain (ETC) and oxidative phosphorylation (OXPHOS) occur to generate ATP. Normal mitochondrial dynamics, including balanced fusion and fission processes, help maintain mitochondrial integrity and prevent fragmentation, swelling, or cristae disruption (reproduced from [32,35]).

**Figure 2 ijms-26-06750-f002:**
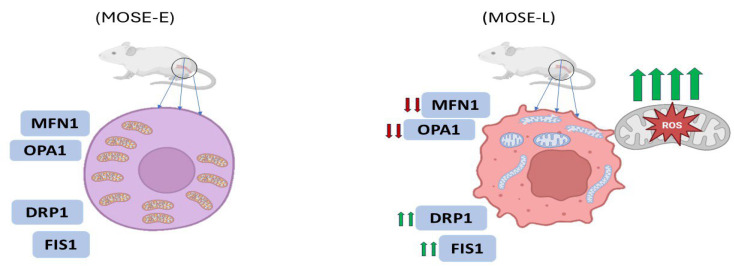
Alterations in mitochondrial ultrastructure during malignant tumor progression. Mitochondrial morphology shifts from an interconnected filamentous network in benign ovarian surface epithelial cells (MOSE-E) to fragmented and enlarged spherical structures in highly aggressive tumor-initiating cells (MOSE-L). These structural changes are associated with dysregulated mitochondrial dynamics, characterized by the downregulation of fusion mediators such as mitofusin 1 (MFN1) and OPA1, and the upregulation of fission-related proteins including DRP1 and FIS1 (reproduced from [34]).

**Figure 3 ijms-26-06750-f003:**
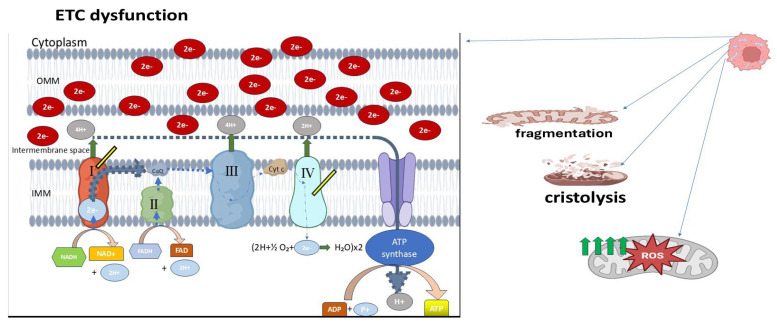
Mitochondrial abnormalities in cancer cells. Tumor cell mitochondria exhibit disorganized cristae, partial cristolysis, and an electron-lucent matrix, indicating structural damage. These mitochondria are highly fragmented and display increased reactive oxygen species (ROS) production due to electron leakage from the electron transport chain (ETC). These dysfunctions contribute to enhanced glycolytic activity, reinforcing the Warburg phenotype commonly observed in cancer metabolism (reproduced from [37,38]).

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
