# Peer review of "Dysregulation of Mitochondrial Function in Cancer Cells"

_ijms, 2025, doi:10.3390/ijms26146750_

Round 1

Reviewer 1 Report

Comments and Suggestions for Authors

In the manuscript by Awad et al., the authors highlighted how cancer cells induce significant reprogramming of mitochondrial structure and function to facilitate their progression.

The manuscript is well written.  I have a few comments for polishing the manuscript.

  1. The authors should add a section highlighting some recent developments of various small molecule inhibitors and drugs that specifically target mitochondria. The authors should review this recent study https://doi.org/10.1016/j.isci.2025.112219 and similar such studies in this respect.
  2. The authors should state the role of oncometabolites in the metabolic rewiring of mitochondrial metabolism in cancers in a more detailed way.
  3. The authors should state the current limitations associated with the research area and what can be done to overcome these limitations.
  4. The authors should state how mitochondrial metabolism contributes to the cancer stem cell phenotype.

Reviewer 2 Report

Comments and Suggestions for Authors

Mitochondria are crucial players in balancing cell survival and death. Indeed, when food supply and oxygen levels are sufficient, they actively and significantly contribute to providing energy in the form of ATP. Conversely, under certain circumstances (e.g., stressor conditions, xenobiotic exposure, etc.), they are "knocked down", thus leading to programmed cell death. Besides these roles, the mitochondria are also pivotal players in the regulation of calcium signalling, various biosynthetic pathways, and are actively involved in producing a large amount of intracellular reactive oxygen species (ROS). In the last dozen years, it has also been established that mitochondria are highly dynamic intracellular compartments. They can fuse and divide, transiently interact with other membranous subcellular compartments, like the Endoplasmic Reticulum (ER), and their number is tightly controlled by a balanced action of biogenesis and removal through mitophagy. Not surprisingly, mitochondrial dysfunction is associated with different disorders, ranging from neurodegenerative diseases to malignancies.

In the manuscript titled "Dysregulation of Mitochondrial Function in Cancer Cells", the authors discuss how cancer cells extensively remodel mitochondrial structure and function, enabling them to survive the hostile tumor microenvironment, evade therapy, and proliferate at a higher rate.

Overall, the manuscript lacks novelty. Many aspects have been neglected. They should have been comprehensively discussed because sort of "milestones" in cancer biology. I refer just to a couple of them: calcium signaling is mentioned but not thoroughly articulated, transient interaction between mitochondria and ER, as well as the metabolomics and oncometabolites alongside IDH mutations and their role in controlling gene transcription are omitted issues. Hence, the manuscript does not offer significant novelty. Before any resubmission, I warmly recommend that the authors carefully consider the reviewer's suggestions to strengthen the manuscript.

Round 2

Reviewer 1 Report

Comments and Suggestions for Authors

In the manuscript by Awad et al., the authors highlighted how cancer cells induce significant reprogramming of mitochondrial structure and function to facilitate their progression.

The authors have addressed all the previous comments. Thus, the manuscript can be accepted in its present form.

Reviewer 2 Report

Comments and Suggestions for Authors

The revised version of the manuscript titled "Dysregulation of Mitochondrial Function in Cancer Cells" has been significantly improved when compared to the original one. The concerns previously raised by the reviewer have been properly addressed; thus, the manuscript has been sufficiently strengthened. Nonetheless, the authors seem to be reluctant to boost it with novelty. I would have appreciated a little bit more initiative and a critical perspective.

Before publication, a few minor amendments are required.

In several parts of the manuscript (e.g., lines 101, 114, 160, 177, 244, 395, 584, etc), the authors detail the publication's year alongside the first author of the study. There is no need; it is redundant because there is already a reference. Please edit accordingly.

In lines 556, 580-581, and 624, instead of the author name and year, replace it simply with the reference number.
